# A Study on OLED Cell Simulation and Detection Phases Based on the A²G Algorithm for Artificial Intelligence Application

Dong-Hun Han [1,†] , Yeong-Hoon Jeong [2,†] and Min-Soo Kang [3,*]

1   Department of Medical Artificial Intelligence, Eulji University, Seongnam 13135, Republic of Korea; d555v@naver.com
2   LG Uplus Corp, Seoul 04389, Republic of Korea; gohawk@lguplus.co.kr
3   Department of Bigdata Medical Convergence, Eulji University, Seongnam 13135, Republic of Korea
*   Correspondence: mskang@eulji.ac.kr
†   Equal first author contributions.

**Abstract:** In this study, we demonstrate the viability of applying artificial intelligence (AI) techniques to conduct inspections at the OLED cell level using simulated data. The implementation of AI technologies necessitates training data, which we addressed by generating an OLED dataset via our proprietary A²G algorithm, integrating the finite element method among concerns over data security. Our A²G algorithm is designed to produce time-dependent datasets and establish threshold conditions for the expansion of dark spots based on OLED parameters and predicted lifespan. We explored the potential integration of AI in the inspection phase, performing cell-based evaluations using three distinct convolutional neural network models. The test results yielded a promising 95% recognition rate when classifying OLED data into pass and fail categories, demonstrating the practical effectiveness of this approach. Through this research, we not only confirmed the feasibility of using simulated OLED data in place of actual data but also highlighted the potential for the automation of manual inspection processes. Furthermore, by introducing OLED defect detection models at the cell level, as opposed to the traditional panel level during inspections, we anticipate higher classification rates and improved yield. This forward-thinking approach underscores significant advancement in OLED inspection processes, indicating a potential shift in industry standards.

**Keywords:** AI; big data; OLED; machine learning

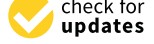



## 1. Introduction

OLED panels, now holding a global market share of over 40%, have gained considerable traction in recent years, predominantly focusing on the development and production of higher-value flexible OLEDs [1]. This rapid expansion and increasing market demand have necessitated stringent and reliable encapsulation processes for these delicate devices. The encapsulation plays a pivotal role in preserving the organic layers from external damage, especially moisture and oxygen infiltration, which is the leading cause of device degradation. Consequently, thin-film encapsulation (TFE) has emerged as the most crucial, albeit demanding task, as OLED devices require the highest degree of protection from moisture and oxygen compared with other electronic devices [2]. Despite employing high-quality materials and advanced encapsulation structures, the organic compounds in OLEDs are highly susceptible to the ravages of moisture and oxygen exposure [3]. This susceptibility often results in irreversible damage, leading to the formation of dark spots—areas where the light emission is markedly impaired. It also precipitates a degradation phenomenon commonly referred to as luminance degradation over time. Even with robust and well-established encapsulation processes, OLEDs invariably experience a certain degree of moisture permeation. This permeation is quantified by the water vapor transmission rate (WVTR), a crucial metric in the evaluation of OLED device longevity and performance. It is

worth noting that moisture permeation in OLED devices is not a unidirectional process, but occurs through various routes. These routes encompass potential reliability issues in the encapsulation method, inherent issues associated with the encapsulation materials used, and manufacturing defects, among others.

To achieve stable OLED operation and enhance device lifespan, it is paramount to maintain WVTR below a certain threshold value. According to the standards set for white OLEDs, WVTR should remain below the specified limit for over 10,000 operational hours [4]. This stringent standard underscores the critical importance of effective encapsulation processes in OLED manufacturing and the continuous need for innovative solutions to improve device durability and performance.

The aforementioned requirement in Table 1 is determined by measuring the time it takes for the magnesium cathode, which serves as an electrode material in OLEDs, to oxidize and become transparent, ultimately transforming into magnesium oxide (MgO). This value was determined by Professor Burrows' team in 2001 [5]. If the OLED components have higher humidity and oxygen permeability than the recommended levels, oxygen and moisture will continuously infiltrate the components during their operating lifetime. This will cause the OLED components to degrade more quickly owing to external factors. To tackle this problem, the International Technology Roadmap for Semiconductors (ITRS) has established a criterion for the largest permissible size of dark spots during the operational lifespan of the display. For an area equivalent to a 300 mm diameter wafer, there should be no more than five small dark spots, each measuring between 20 and 30 nanometers in size. Major manufacturers of OLED panels, such as Samsung Display and LG Display, have even more stringent requirements for mass production. However, the specifics of these standards are confidential and cannot be disclosed.

**Table 1.** Encapsulation requirements.

| Parameter | Minimum Requirements |
|---|---|
| Water vapor transmission rate | $10^{-6}$ g/m$^2$-day |
| Oxygen permeability rate | $10^{-5}$ cm$^3$ (STP)/m$^2$-day |
| Dark spot standard | 300 mm/under 5 dark spots |
| Wafer | 20~30 nm |

To detect defects using machine learning, it is necessary to train the model on a large amount of data. However, because of security concerns in the OLED industry, obtaining a substantial amount of data is challenging. Therefore, in this study, we generated a simulated dataset using the finite element method to detect defects using artificial intelligence.

The deep convolutional neural network (CNN) is the state-of-the-art solution for large-scale visual recognition [6,7]. To assess the application of artificial intelligence, we conducted recognition rate tests using representative algorithms, namely, VGG-16, ResNet50, and Conv-Next Tiny. To evaluate the prediction accuracy, we compared metrics such as accuracy, loss, AUC-ROC, specificity, and sensitivity among the algorithms. The final confirmation was made by assessing the accuracy of predictions based on these metrics.

## 2. Related Research

### 2.1. Moisture Diffusion for OLED Artificial Intelligence Applications

In the field of OLED displays, artificial intelligence has been continuously used to research and develop new materials and substances. AI has been employed in the process of synthesizing and evaluating molecular structures to achieve desired properties. Recently, there have been ongoing efforts to apply AI in inspection stages.

Many OLED manufacturers grapple with low yield rates due to defects such as dark points, surface scratches, non-uniform luminance, and lack of color uniformity [8]. Broadly speaking, these defects can be divided into two groups. One is glass defects and the other is cell defects. Defects can manifest in the form of scratches, contamination, hair, and dust particles. Scratches and other similar defects are considered genuine defects, while

defects that can be removed by air, such as hair and dust, are referred to as superficial defects. Research on artificial intelligence for data-driven inspection processes to detect genuine cell defects is currently underway based on studies of dark spot growth. The paper titled "Extraction of the OLED Device Parameter based on Randomly Generated Monte Carlo Simulation with Deep Learning" by Seung Yeol You and two other authors proposes techniques for accelerated computer simulation and parameter reverse engineering [9]. These techniques efficiently analyze the relationship between OLED process variables and luminescent performance.

Dark spots can be influenced by various factors including the thickness of the organic light emitting layer (L), humidity ($C_0$), temperature (T), pressure (P), and degradation time (t) [10]. Conducting performance experiments based on all process conditions can be very costly. Therefore, computer simulations were generated using each factor to obtain results. The generated samples represent the growth of dark spots with varying degradation times (t), number of pinholes (N), and sizes ($r_0$). The growth of dark spots can be represented by Equation (1), which relates the radius (D) and degradation time (t) [11].

$$dw/dt = (\beta LDC_0)/(4\pi L/r_0) \cdot 1/d \tag{1}$$

where d = organic layer diffusion coefficient, w = diffusion area, t = degradation time, $\beta$ = water flux per unit coefficient, L = thickness of EL, $r_0$ = pinhole size, $C_0$ = humid, and D = dark spot area.

Equation (1) is similar to the one presented in the paper titled "Dark spot growth and its acceleration factor in Organic Light Emitting Diodes with Single Barrier Structures" by Takeru Okada and his colleagues [12]. This equation is based on the comparison of the performance of organic light emitting diodes (OLEDs) under high temperature and high humidity to that of edge shrinkage at the cathode edge.

In the case of the diffusion of dark spots, a moisture diffusion equation based on pinholes is essential. The equation for moisture diffusion can be defined as follows: The water flux per unit length is $LDC0 = w$ when the diffusion coefficient of the organic layer is D and the thickness of the organic layer is L. It is assumed that D is a function of temperature but is independent of water concentration within the range of this experiment. Assuming that the rate of edge shrinkage is a product of the water flux per unit length and the coefficient $\beta$, the width of the edge shrinkage, w, can be expressed in a rate equation.

$$dw/dt = (\beta LDC_0)/w \tag{2}$$

Solving this equation leads to

$$w = \sqrt{(2\beta LDC_0 \, t.)} \tag{3}$$

Based on this equation, a computer simulation for the dataset of dark spot creation and diffusion was conducted.

### 2.2. Impact of Dark Spots on OLED and the Use of Artificial Intelligence

In a research paper titled "Effect of Organic Layer Combination on Dark Spot Formation in Organic Light Emitting Devices" authored by Yoon-Fei Liew et al., the evolution of dark spots in OLEDs was examined through optical image analysis technology [13]. The study unearthed that both single and multilayered OLEDs can develop dark spots, negatively influencing the performance of OLEDs' organic constituents. During the operational degradation of OLEDs, non-emissive "dark spots" can emerge, with potential causes being manufacturing defects and damage to surface protection [14]. The insights garnered from this paper lead to the understanding that the organic layers integral to OLEDs contribute to the formation of dark spots.

Therefore, the paper's primary focus lies in investigating the morphology of dark spots and enhancing the lifespan and efficiency of OLED displays by understanding the implications of dark spots on OLEDs.

Currently, automated optical inspection (AOI) is commonly used in production to detect pre-defined Mura, providing accurate and fast results compared with human expertise [15]. However, AOI has a limitation as it cannot detect new Mura [16,17]. Another approach for defect detection is through a vision system, often employed for position-based defect detection of panels. This method requires high precision inspection equipment, and errors can lead to significant costs [18]. In the paper titled "A Study on Defect Detection of Flexible OLED Using Deep Learning" by Kim, S. K. et al., the researchers used a deep learning algorithm to identify defects in flexible OLEDs [19]. After learning about defects in OLEDs, a deep learning algorithm was applied. The test results showed high performance in detecting not only previously learned defects but also multiple defects [20]. In this study, machine learning was used to detect Mura based on the panel, in contrast to the current research that focused on the cell unit level and used different artificial intelligence algorithms for detecting dark spots and predicting lifespan.

Based on the case study above, it is confirmed that dark spots can harm OLED cells and ultimately cause Mura expansion. Therefore, dark spots play an important role in predicting the lifespan of OLEDs. Several studies have explored the use of artificial intelligence in inspection processes. However, this study is unique in its focus on cell-level analysis rather than panel-based inspection.

### 3. Research on OLED Prediction Based on Dark Spots

Dark spots in OLEDs can be influenced by various factors, including the thickness of organic light emitting layer, humidity, temperature, pressure, and aging time. To implement a dataset using the finite element method (FEM), we used an FEM solver and identified factors that directly or indirectly affect the performance of OLED cells as follows.

Pinhole size and number were determined based on a study titled "Extraction of the OLED Device Parameter based on Randomly Generated Monte Carlo Simulation with Deep Learning" by Seung Yeol You et al. Criteria for image processing in big data were established using minimum moisture content as a factor [10]. Temperature and humidity were maintained at 25 °C and 50%, respectively, to ensure consistency in the dataset, while the atmospheric pressure was set to 1 atm, which is the standard for ambient air. All datasets were created with a minimum requirement of 10,000 h of lifetime, which meets the minimum requirement for the lifetime of the white OLED. The diffusion coefficient can be calculated using the average free path and average velocity of molecules in an ideal gas distribution, such as the Maxwell–Boltzmann distribution. One of the goals of this research, namely the generation of the OLED dataset, involved constructing a three-dimensional composition of the continuously flowing time and space variables in order to calculate the diffusion of moisture spaces based on the flow of 10,000 h and pinholes, as depicted in Figure 1.

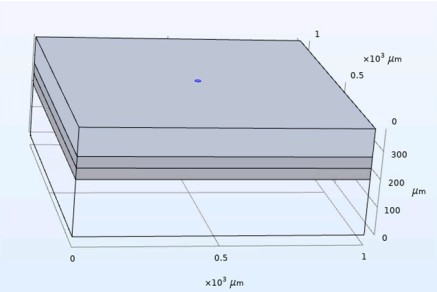

**Figure 1.** OLED cell 3D simulated.

OLEDs encompass various components such as organic resonant tunneling diodes, organic phototransistors, organic photodetectors, and organic photovoltaic cells [21]. Thus, an OLED is made up of several layers composed of organic matter sandwiched between two electrodes, an anode and a cathode [22]. Figure 1 demonstrates the simulation process of organic light-emitting diodes (OLEDs). This process is fundamentally rooted in the effects of surface chemical treatments on the morphology of inkjet-printed polymer films [23], highlighting that pinholes, an integral part of injecting organics into each layer, are a natural consequence of this process. Further to this, as depicted in Figure 2, we finely segmented the area where the dark spots proliferate by methodically slicing it layer by layer and subdividing it into finite meshes.

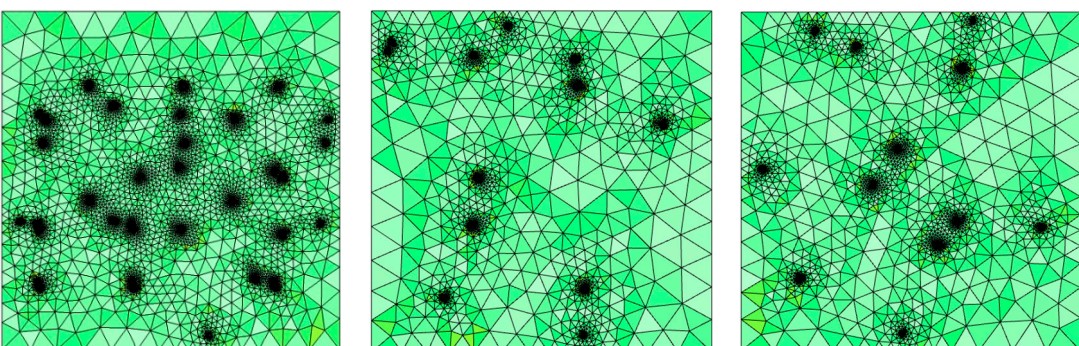

**Figure 2.** OLED dark spot simulated cell based on FEM.

This divided mesh serves as a measure of the precision achieved during the physical simulation within each internal area. The areas colored deeper green indicate higher accuracy in the results. All datasets were processed with mesh precision exceeding 75%. Moreover, because of denser meshing, higher simulation accuracy, and the longer computational time [24], employing a 'Fine Mesh' with heightened precision ensured that each segment had a minimum of 100 mesh values.

The FEM Solver used was COMSOL Multiphysics Simulation. COMSOL Multiphysics® (known as FEMLAB before 2005) is a commercial finite element software package designed to address various physical phenomena [25,26]. The solver calculates the area of moisture spread from pinholes in the EL liquid layer of the OLED display using finite values. In the case of dark spots, their formation is based on the work of Ohzu et al., where they calculated the dark spot formation in flexible OLEDs in relation to the amount of water that penetrated into the device [27]. Additionally, the integrity of the finite element method was verified through mesh analysis, which is one of the validation techniques used in FEM. The image dataset was created using our own algorithm called OLED A$^2$G (Automatic Affine Generator). Algorithm 1 is designed as follows.

OLED A$^2$G is a sophisticated, time-dependent dataset generation algorithm that we developed with a specific focus on creating datasets for the diffusion of moisture, particularly as it pertains to pinholes—a significant factor affecting the lifespan of OLED displays. This algorithm expertly captures and predicts essential information about OLED lifespan, which is accomplished by closely examining and modeling the occurrence and expansion of pinholes. This process is accomplished through a combination of computational modeling and advanced data processing techniques. In this research, we determined the number of datasets to be generated through the variable known as oled_data_count. For simplicity and compatibility, the file extension was specified as .png. We used the variable pinhole_count to represent the number of pinholes created during the computational simulation process. The variables OLED_cell_x and OLED_cell_y were designated to indicate the size of cells used in the computational model. Furthermore, to account for the natural variation that occurs in the manufacturing process, the range and the number of pinholes were randomly generated using base and m_base variables. Upon generating these pinholes within the model, they were then incorporated into the cell. The time frame from 0 to 10,000 h was

divided into sequential frames and computed according to the specified Equation (3). The dataset generated through this rigorous process is subsequently used in machine learning algorithms to determine the lifespan and defect classification criteria of OLED panels. While these data may not be directly applicable to an artificial intelligence model, they nonetheless lay a solid foundation for such a model by producing a significant volume of computationally simulated data.

---

**Algorithm 1** OLED A$^2$G (Automatic Affine Generator)

---

```
# Feature
oled_data_count: Number
File_extention: PNG
Pinhole_count: Number

# Variable
x: oledcell x axis
y: oledcell y axis
base: pinhole x
m_base: pinhole y
base, m_base: random number in oledcell x and y
# Study
WHILE oled_data_count > 0:
  SET COMPONENT x, y to base, m_base
    Comsol build mesh
    IF mesh_accuracy >= 80%:
      Comsol build time_dependent
      Comsol build frame
      run molecule diffusion study
      save data in "location"
    ELSE:
      rebuild
oled_data_count -= 1
```

---

After completing this large image dataset, we conducted a thorough accuracy analysis among several machine learning algorithms. The aim of this analysis was to compare the performance of various prediction models when they were exposed to the same dataset. As a result of this analysis, we identified the model with the highest accuracy. The prepared OLED dataset was then fed into this top-performing model to evaluate its recognition rate. This intricate process plays a vital role in enhancing the practicality and effectiveness of AI applications in the field of OLED inspection and prediction.

### 4. Discussion

A comprehensive dataset of 2000 image samples was generated for the study. These images were created by specifying the grayscale color table within the A$^2$G algorithm, based on the parameter values outlined in Table 2. To ensure suitable threshold determination, the threshold level was established according to minimum moisture content criteria. The minimum moisture level criterion was defined based on the observation that, when the moisture within the OLED substrate reaches a certain threshold, the OLED luminance decreases by more than 50%. Consequently, the images generated visually illustrate the spread of moisture, represented by randomly distributed dark spots rendered using grayscale. The generated dataset was then applied to various machine learning algorithms to assess their performance, and the comparative results are summarized in Table 3. In-depth analysis of the results presented in the form of graphical representations can be found in Figure 3. Through these rigorous evaluations and comparisons, the performance of each machine learning algorithm in predicting moisture spread, as represented by dark spot distribution, was assessed. The assessments were aimed at identifying the most effective algorithm for predicting and analyzing moisture distribution and its effects on OLED performance.

**Table 2.** Simulation parameter.

| Parameter | Range |
|---|---|
| Initial pinhole radius | 1.2 (µm) |
| Number of dark spots | 10~100 (Ea) |
| Degradation time | 10,000 (h) |
| Minimum humidity inflow (for edge detection) | 1 (%) |
| OLED panel width/length | 1000/1000 (µm) |
| Moisture | 50% |
| Temperature | 25 (°C) |
| Absolute pressure | 1 ATM |
| diffusion coefficient | $2.6 \times 10^{-5}$ [µm$^2$/s] |
| Max dark spot growth | 10% |

**Table 3.** Machine learning results.

| Algorithm | Loss | Accuracy | Specificity | Sensitivity | AUC-ROC |
|---|---|---|---|---|---|
| VGG-16 | 0.053 | 0.982 | 0.981 | 0.985 | 0.983 |
| ResNet50 | 0.052 | 0.981 | 0.98 | 0.984 | 0.982 |
| Conv-Next Tiny | 0.039 | 0.986 | 0.988 | 0.984 | 0.986 |

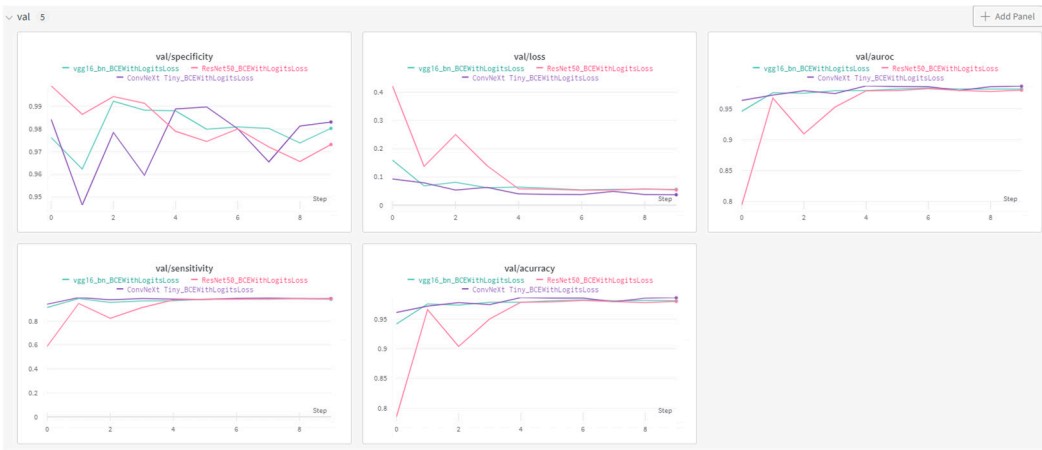

**Figure 3.** Machine learning algorithm results graph.

From the results displayed in Table 3 and Figure 3, it is evident that the Conv-Next Tiny algorithm emerges as a superior performer among its counterparts, exhibiting the lowest loss value and the highest accuracy. These results strongly suggest that the Conv-Next Tiny algorithm possesses an advanced ability to learn and recognize complex dark spot patterns with a high degree of precision. This is a critical attribute, especially considering the intricate structures and behaviors of OLEDs, which demand meticulous handling in analysis and classification tasks.

While Conv-Next Tiny outperforms other algorithms, it is equally important to highlight that other convolutional neural network (CNN) algorithms, such as VGG-16 and ResNet50, also deliver impressive accuracy values of 0.98 or above. This not only underscores the overall effectiveness of CNN-based algorithms in defect detection tasks, but also indicates their proficiency in identifying particular anomalies such as dark spots. As we delve deeper into a more nuanced analysis, considering the unique features and characteristics of each algorithm, it becomes clear that lightweight algorithms such as Conv-Next Tiny offer certain distinct advantages. For instance, these algorithms provide faster training time and improved computational performance compared with their more complex counterparts. This is a crucial factor in practical situations where the speed of execution and efficient utilization of computational resources are of utmost importance.

A testament to the practicality and effectiveness of the Conv-Next Tiny algorithm is its superior performance on a batch of 100 OLED test data samples, which were classified as either pass or fail. The Conv-Next Tiny model, which boasted the highest accuracy among the trained algorithms, achieved a remarkable recognition rate of 95%. This excellent performance in real-world testing strongly suggests that the Conv-Next Tiny algorithm has great potential to serve as a powerful tool for accurate and efficient OLED defect detection.

Such proficiency in defect detection can greatly benefit OLED manufacturing processes by potentially reducing wastage and improving yield. The significance of this can be further appreciated considering the economic and environmental implications of waste reduction in the manufacturing industry. This breakthrough underscores the potential of machine learning algorithms, particularly lightweight models such as Conv-Next Tiny, in revolutionizing OLED manufacturing processes and paving the way for industry-wide adoption of AI-based inspection and prediction techniques.

## 5. Conclusions

OLEDs, known for their requirement of sustained brightness over prolonged periods, pose challenges when traditional physical methods of measurement are employed because of their inherent limitations. Recognizing these challenges, our study showcases the promising potential of using $A^2G$-based simulated data generation and artificial intelligence techniques as alternatives for longevity measurements and passive inspections of OLEDs. To combat the scarcity of training data, a dataset revolving around dark spots, which significantly affect the OLED lifespan, was synthesized using the finite element method (FEM) and the $A^2G$ algorithm. This synthesized dataset served as the foundation to train an AI model. Remarkably, the trained AI model demonstrated high proficiency in recognizing OLED test data, hinting at its valuable application in defining the criteria for defective OLED panels. These findings underline the possibility of automation in the inspection process and project the capability of executing more precise AI inspections at the intricate OLED cell level, shifting away from conventional panel-level inspections. Furthermore, the $A^2G$ algorithm, which is employed for creating OLED simulation data, enables result confirmation up to 10,000 h through data generation via virtualization post-physical calculation-based design. This methodology deviates from the traditional sequential design–manufacture–evaluation process, highlighting the feasibility of assessing various operational results within a significantly reduced time frame.

Considering these findings, we advocate for the expansion of future research to delve into potential issues that may arise within the OLED manufacturing process. Incorporating AI throughout the complete process, including the inspection stage, opens up the possibility of crafting research that considers the impact of factors such as organic materials, plastics, and rubbers that occur within the various OLED layers. This holistic approach can lead to better quality control and efficiency in OLED production, advancing the technology to new heights.

Moreover, this work suggests a roadmap toward the advent of an integrated manufacturing process underpinned by artificial intelligence. Future studies can explore and expand the application of AI models and algorithms not only in OLED manufacturing, but also in other microelectronic devices. Additionally, it would be worth investigating the incorporation of AI-driven feedback mechanisms that can self-improve and self-correct during the manufacturing process. The combination of AI-driven data generation, AI-based predictive modeling, and AI-enabled decision-making and control mechanisms could usher in a new era of smart manufacturing, contributing to the Fourth Industrial Revolution. We also recommend the development of standard testing protocols that can fully exploit the advantages of AI, thereby ensuring reliable and robust manufacturing processes. By using AI in tandem with traditional manufacturing practices, we foresee a future where technology and human expertise coalesce, leading to remarkable advancements in the electronics industry.

**Author Contributions:** D.-H.H. and Y.-H.J.: Writing—original draft, Data curation and software, Visualization, Data curation. M.-S.K.: Conceptualization, Validation, Writing—review and editing, Project administration. Y.-H.J.: Formal analysis, Funding acquisition. Equal first author contributions. All authors have read and agreed to the published version of the manuscript.

**Funding:** This research received no external funding.

**Institutional Review Board Statement:** Not applicable.

**Informed Consent Statement:** Not applicable.

**Data Availability Statement:** All data, models, and code generated or used during the study are available from the corresponding author by request.

**Conflicts of Interest:** The authors declare no conflict of interest.

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
