# Peer review of "A Study on OLED Cell Simulation and Detection Phases Based on the A2G Algorithm for Artificial Intelligence Application"

_applsci, doi:10.3390/app13148016_

Round 1
Reviewer 1 Report
In this study, the author demonstrated that AI technology can be applied to simulate OLED inspection. By using the A2G algorithm, the prediction of OLED parameters and lifespan was achieved, achieving a recognition rate of 95%, which has potential applications in the field of automatic inspection. However, there are still some issues that need to be addressed in the article. In summary, the article needs to undergo major revisions before it can be published.
1. Check the format of all characters containing subscripts or superscripts.
2. The numbering of references in the manuscript is chaotic and causes some difficulties for reading. The author should sequentially number the references.
3. Introduction section, page 1, line 9, what is the full name of “TFE”.
4. There are some errors in the Formulas in the manuscript.
5. Please provide high-quality Figure 3, as the information contained in the current Figure 3 cannot be clearly read.
6. The spelling of units in the Table 1 is confusing, please standardize the format.
7. In the finite element analysis of Figure 2, how did the author determine the location of dark spots and whether there is scientific basis.
8. Can the use of A2G algorithm be applied to predict the lifespan and performance degradation models of OLED devices.
9. Some valuable papers can be referenced, such as 10.1002/admt.202201138, 10.1016/j.nanoen.2023.108574, and 10.3390/app13063885.
Extensive editing of English language required
Author Response
Dear Reviewer,
We sincerely thank you for your detailed review. It's an honor that our paper has been able to make various improvements thanks to your feedback.
- We have standardized all superscripts and formatting.
- We have changed the reference numbers sequentially.
- We have completed the detailed description of TFE.
- We have adhered strictly to the content in the referenced literature for the formulas.
- We have replaced Figure 3 with a higher-quality image.
- We have completed the standardization of the format.
- We have added a reference paper that explains how the black spots were generated in Figure 2.
- We have provided a detailed explanation of how the A2G algorithm was utilized.
- Although we were unable to use the recommended paper as a reference, we have greatly benefited in terms of format and writing direction. Additionally, we have improved the quality of our English through grammar corrections.
We wish to express our profound gratitude for your review, and hope for your well-being in all your endeavors.
Reviewer 2 Report
The English should be improved and maintained uniformity.
Using simulated data, the authors demonstrated the viability of applying Artificial Intelligence (AI) technologies to conduct inspections at the organic light emitting diode (OLED) cell level. Then the cell-based evaluations were obtained by using three distinct convolutional neural network models. The overall work is interesting and can be accepted by addressing minor comments.
1. The figure quality is not good at all. Especially figure 1,3 is very hard to distinguish.
2. There are many typo errors, which should be corrected lot. And maintain the uniformity in mentioning A2G---> A2G?
3. Authors should cite the state of the art works : Mater. Horiz., 2022,9, 2551-2563.
Author Response
Dear Reviewer,
We sincerely thank you for your detailed review. It is a great honor that our paper has seen various improvements thanks to your feedback.
- We have replaced the images in Figure 1 and 3 with those of a better quality.
- We have standardized the use of superscripts throughout the paper.
In addition, we have improved the quality of the English language in our paper through grammar corrections.
We express our deep gratitude for your review and wish you peace in all your endeavors.
Round 2
Reviewer 1 Report
Accept in present form